# Anatomical and functional connectivity support the existence of a salience network node within the caudal ventrolateral prefrontal cortex

Lucas R Trambaiolli[1,2†], Xiaolong Peng[3,4†], Julia F Lehman[2], Gary Linn[5], Brian E Russ[5,6,7], Charles E Schroeder[5,8], Hesheng Liu[3,4†], Suzanne N Haber[1,2*†]

[1]McLean Hospital, Harvard Medical School, Belmont, United States; [2]University of Rochester School of Medicine & Dentistry, Rochester, United States; [3]Massachusetts General Hospital, Harvard Medical School, Boston, United States; [4]Medical University of South Carolina, Charleston, United States; [5]Translational Neuropscienc lab Division, Center for Biomedical Imaging and Neuromodulation, Nathan S. Kline Institute for Psychiatric Research, Orangeburg, United States; [6]Nash Family Department of Neuroscience and Friedman Brain Institute, Icahn School of Medicine at Mount Sinai, New York, United States; [7]Department of Psychiatry, New York University at Langone, New York, United States; [8]Department of Psychiatry, Columbia University Medical Center, New York, United States

**\*For correspondence:**
Suzanne_Haber@urmc.rochester.edu

[†]These authors contributed equally to this work

**Competing interest:** The authors declare that no competing interests exist.

**Abstract** Three large-scale networks are considered essential to cognitive flexibility: the ventral and dorsal attention (VANet and DANet) and salience (SNet) networks. The ventrolateral prefrontal cortex (vlPFC) is a known component of the VANet and DANet, but there is a gap in the current knowledge regarding its involvement in the SNet. Herein, we used a translational and multimodal approach to demonstrate the existence of a SNet node within the vlPFC. First, we used tract-tracing methods in non-human primates (NHP) to quantify the anatomical connectivity strength between different vlPFC areas and the frontal and insular cortices. The strongest connections were with the dorsal anterior cingulate cortex (dACC) and anterior insula (AI) – the main cortical SNet nodes. These inputs converged in the caudal area 47/12, an area that has strong projections to subcortical structures associated with the SNet. Second, we used resting-state functional MRI (rsfMRI) in NHP data to validate this SNet node. Third, we used rsfMRI in the human to identify a homologous caudal 47/12 region that also showed strong connections with the SNet cortical nodes. Taken together, these data confirm a SNet node in the vlPFC, demonstrating that the vlPFC contains nodes for all three cognitive networks: VANet, DANet, and SNet. Thus, the vlPFC is in a position to switch between these three networks, pointing to its key role as an attentional hub. Its additional connections to the orbitofrontal, dorsolateral, and premotor cortices, place the vlPFC at the center for switching behaviors based on environmental stimuli, computing value, and cognitive control.

## Editor's evaluation

This is an interesting quantitative study of the anatomical connections of a region of prefrontal cortex that has often been overlooked because it is at the border of what is typically called ventro-lateral prefrontal cortex and orbitofrontal prefrontal cortex. Sometimes it is included as part of ventrolateral prefrontal cortex, sometimes as part of orbitofrontal cortex and sometimes it is simply given little attention because ventrolateral studies focus on the inferior convexity and orbital studies

focus on the region between the orbitofrontal sulci. The idea that this is a special region that is different from both the rest of ventrolateral prefrontal cortex and probably the rest of orbitofrontal cortex is important because it helps us understand some otherwise puzzling results. The quantitative analysis of connections was an unusual strength of the study as was the comparison of tracer data in macaques, fMRI connectivity data in macaques, and human fMRI connectivity data.

## Introduction

Three distributed attentional networks, the dorsal and ventral attention (DANet and VANet) and salience (SNet) networks, play key roles in switching actions based on environmental stimuli (*Knudsen, 2007*; *Corbetta et al., 2008*; *Seeley et al., 2007*). The DANet is a top-down bilateral fronto-parietal network, responsible for *selecting* stimuli and responses (*Corbetta et al., 2008*; *Corbetta and Shulman, 2002*). The VANet is a bottom-up ventral fronto-parieto-temporal network, responsible for *detecting* outstanding stimuli and reorienting ongoing activity (*Corbetta et al., 2008*; *Corbetta and Shulman, 2002*). The salience network (SNet) (*Seeley et al., 2007*; *Uddin, 2016*), cortically anchored in the anterior insula (AI) and the dorsal anterior cingulate cortex (dACC), adds value to external and internal stimuli, driving attention to rapidly modify behaviors (*Seeley, 2019*). The SNet works closely with the VANet, to 'pull' attention to valued stimuli, based on a combination of previous experience and motivation. However, all three networks must operate together for rapid environmental responses. The ventrolateral prefrontal cortex (vlPFC) lies at the junction between the DANet (areas 44 and 45) (*Rossi et al., 2007*; *Wardak et al., 2010*; *Kadohisa et al., 2015*; *Bichot et al., 2015*; *Bichot et al., 2019*; *Hartwigsen et al., 2019*; *Buckner et al., 2011*) and VANet (area 47/12) (*Hartwigsen et al., 2019*; *Buckner et al., 2011*; *Romanski, 2007*; *Kar and DiCarlo, 2021*; *Romanski and Chafee, 2021*). In contrast, based on imaging studies, the key nodes of the SNet are ACC and AI, and not the vlPFC. Yet, the vlPFC, particularly area 47/12, is central for assessing value and, along with the ACC drives information seeking, to provide value-related discriminations (*Monosov and Rushworth, 2022*). Indeed, it is the orbito-lateral portion of area 47/12 that is involved in stimulus-outcome predictions (*Rudebeck et al., 2017*; *Grohn et al., 2020*; *Jezzini et al., 2021*), and, when lesioned, interferes with choices based on outcome availability (*Rudebeck et al., 2017*). Area 47/12 is tightly connected to both the ACC and the adjacent AI (*Petrides and Pandya, 2002*). However, area 47/12 is large and connected to a wide range of cortical regions. We posit that embedded within this large area is a separate SNet node that links the ACC and AI with the vlPFC that has not been evident due to the technical limitation of functional MRI (*Seeley et al., 2007*; *Seeley, 2019*; *Sridharan et al., 2008*). We demonstrate here, that, based on its anatomic organization and connections to the two central nodes of the SNet (dACC and AI) the vlPFC is a distinct node in the SNet, separate from the adjacent AI. We also show that, with anatomic guidance, this separate node can be identified using fMRI in the human brain. A SNet component within area 47/12 brings unique information about stimulus value to this network, through its connections with the orbitofrontal cortex and thus complementary to the roles of the AI and dACC in information integration and information seeking, respectively. Given the high interconnectivity of areas 44, 45, and 47/12, a SNet node within the vlPFC places it in a central hub-like position to integrate information across the three main attention networks, supporting the region's central role in modulating behavioral flexibility (*Dajani and Uddin, 2015*; *Badre and Wagner, 2006*; *Waegeman et al., 2014*).

We used a cross-species and cross-modality approach to determine the relative strengths of connections of subregions of the vlPFC with the two SNet cortical nodes, the AI and ACC, compared to other frontal regions: tract-tracing methods in macaque monkeys, followed by a seed-based fMRI approach to determine connectivity strength first in the NHP then in humans. We first quantified the anatomic connectivity strength between the different vlPFC subregions and the frontal and insular cortices. We found that the strongest connections with the dACC and AI were with the caudal area 47/12. This sublocation also presented strong axonal projections to subcortical structures of the salience network, including the dorsomedial thalamus (DT), sublenticular extended amygdala (SEA), substantia nigra/ventral tegmental area (SN/VTA), and periaqueductal gray (PAG). Using resting-state functional connectivity MRI (fcMRI), we found that the connectivity strength and patterns between the subregions of the vlPFC and the dACC and AI SNet nodes were similar to anatomic data in NHP. Finally, placing seeds in homologous vlPFC regions in the human, we show that, similar to the

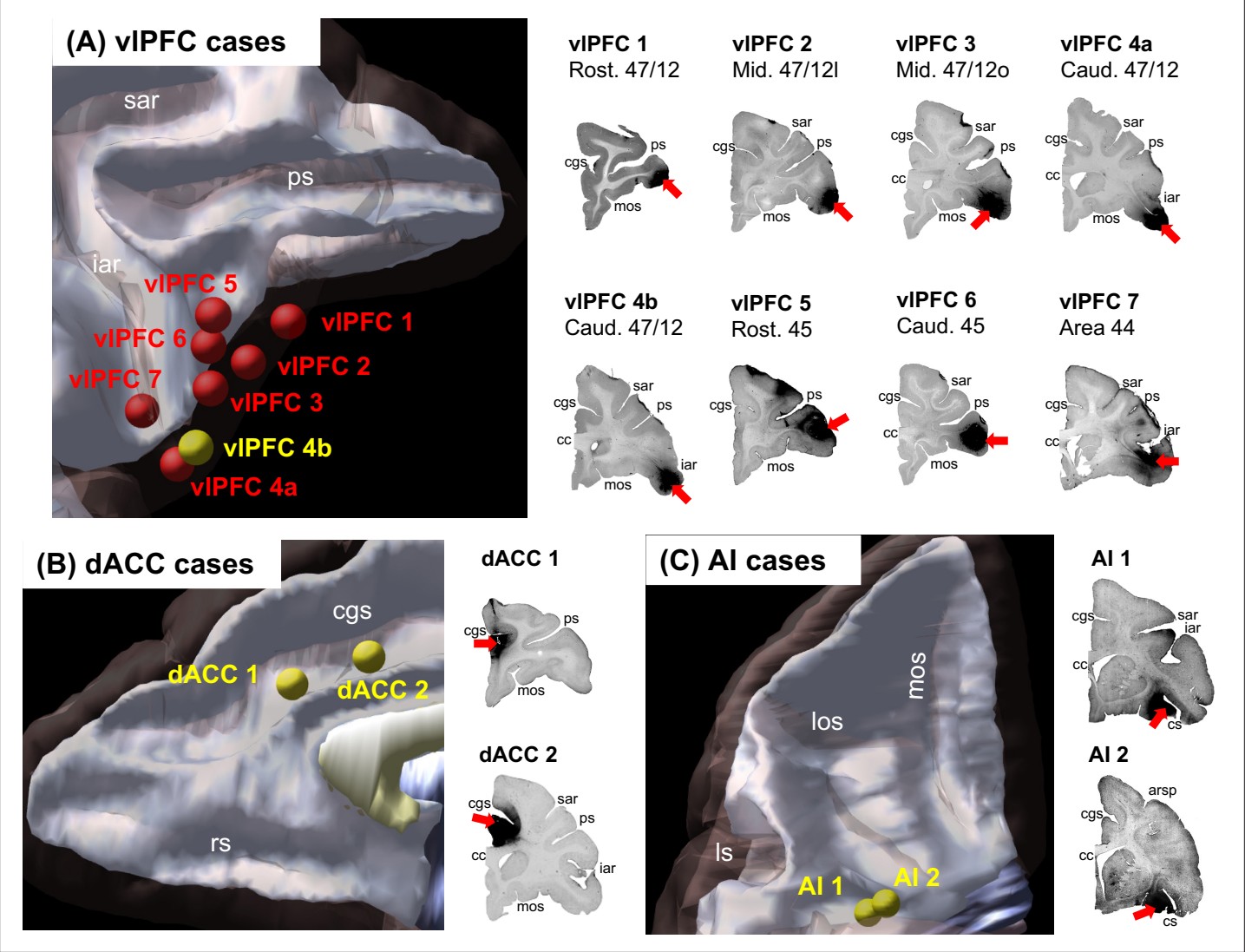

**Figure 1.** Injection sites. (**A**) Location of 8 injection locations in the vlPFC selected for retrograde analysis. Seven cases were analyzed as the main results (red), and one case was used as validation (yellow). Injection locations in (**B**) the dACC and (**C**) the AI selected for anterograde validation of the salience node. *Abbreviations:* arsp = arcuate sulcus spur; cc = corpus callosum; cgs = cingulate sulcus; cs = circular sulcus; iar = inferior arcuate sulcus; ls = lateral sulcus; los = lateral orbital sulcus; mos = medial orbital sulcus; ps = principal sulcus; rs = rostral sulcus; sar = superior arcuate sulcus.

NHP results, fcMRI connectivity between caudal 47/12 is significantly stronger with the dACC and AI compared to other vlPFC regions.

## Results
### Retrograde tracing reveals a SNet node in the caudal area 47/12

Retrograde tracing injections were placed in areas 44, 45 and subregions of 47/12 on the right vlPFC (coronal representations of injection centers and 3D view of injections in *Figure 1*) and the labeled cells in the frontal and insular cortices were charted. We focused on the right hemisphere to reduce the effect of species specificities associated to language development in our analyses (*Nozari and Thompson-Schill, 2016*). To determine the relative projection strengths across cases, we calculated the percentage of total labeled cells that project from each cytoarchitectonic area to each injection site. To compare the projection strengths to what would be expected by chance, we performed a random sampling analysis by permuting neurons $10^6$ times among each frontal or insular cortex area with a probability given by the volume of each area. To evaluate the strength of connections from

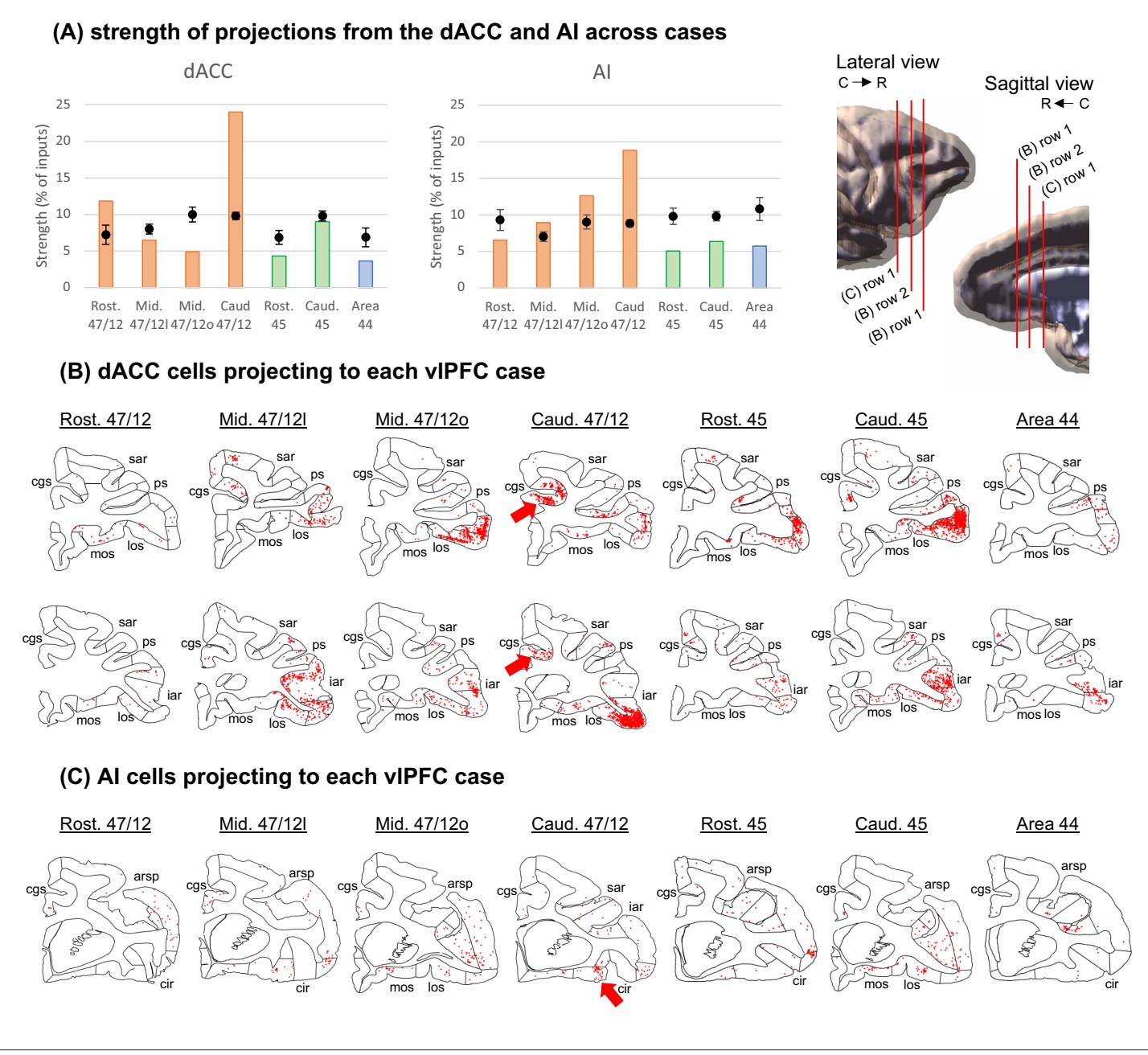

**Figure 2.** Strength of projections from Salience Network cortical nodes grouped by cytoarchitectonic divisions across cases. (**A**) The dACC corresponds to area 24, while the AI is the combination of areas OPAl, Opro, IPro, and AI. Orange bars illustrate cases with injections in area 47/12, green bars in area 45, and blue bars in area 44. Black dots show the average and standard-deviation of random sampling from the respective areas in each case. 3D models represent the location of coronal slices from figures B-C. Coronal sections and the respective labeled cells (red dots) in the (**B**) dACC and (**C**) AI projecting to the caudal area 47/12 in the vlPFC. The black circles represent the areas of interest for the Salience Network. *Abbreviations:* arsp = arcuate sulcus spur; cgs = cingulate sulcus; cir = circular sulcus; iar = inferior arcuate sulcus; los = lateral orbital sulcus; mos = medial orbital sulcus; ps = principal sulcus; sar = superior arcuate sulcus.

The online version of this article includes the following figure supplement(s) for figure 2:

**Figure supplement 1.** Strength of projections from the frontal and insular cortices to different regions of the vlPFC.

**Figure supplement 2.** Labeled input neurons following retrograde tracer injections in different vlPFC locations.

the main cortical nodes of the SNet, we compared projections from the dACC (area 24) and AI (areas OPAl, OPro, IPro and AI) across cases. The results demonstrate that the connectivity strength varies across vlPFC areas (*Figure 2A*, extended bar charts are shown in *Figure 2—figure supplement 1*).

Among all vlPFC injections, caudal area 47/12 stands out as the main location for connections from the dACC and the AI. This area, in addition to rostral 47/12, showed connectivity strength above the chance level with dACC (area 24). Specifically, area 24 projections to caudal area 47/12 were at least twice as strong as expected by chance and twice as strong compared to the projections to the other vlPFC locations. Clusters of projecting cells were found in both pre- and post-genual dACC (*Figure 2B*) in a rostrocaudal distribution consistent with the SNet description in NHP (*Touroutoglou et al., 2016*). For projections from the AI, caudal area 47/12 had the highest difference from the chance level, twice as high compared with injections in mid 47/12. Interestingly, these cells clusters are located in the orbital portion around the beginning of the circular sulcus in the AI. Specifically, this

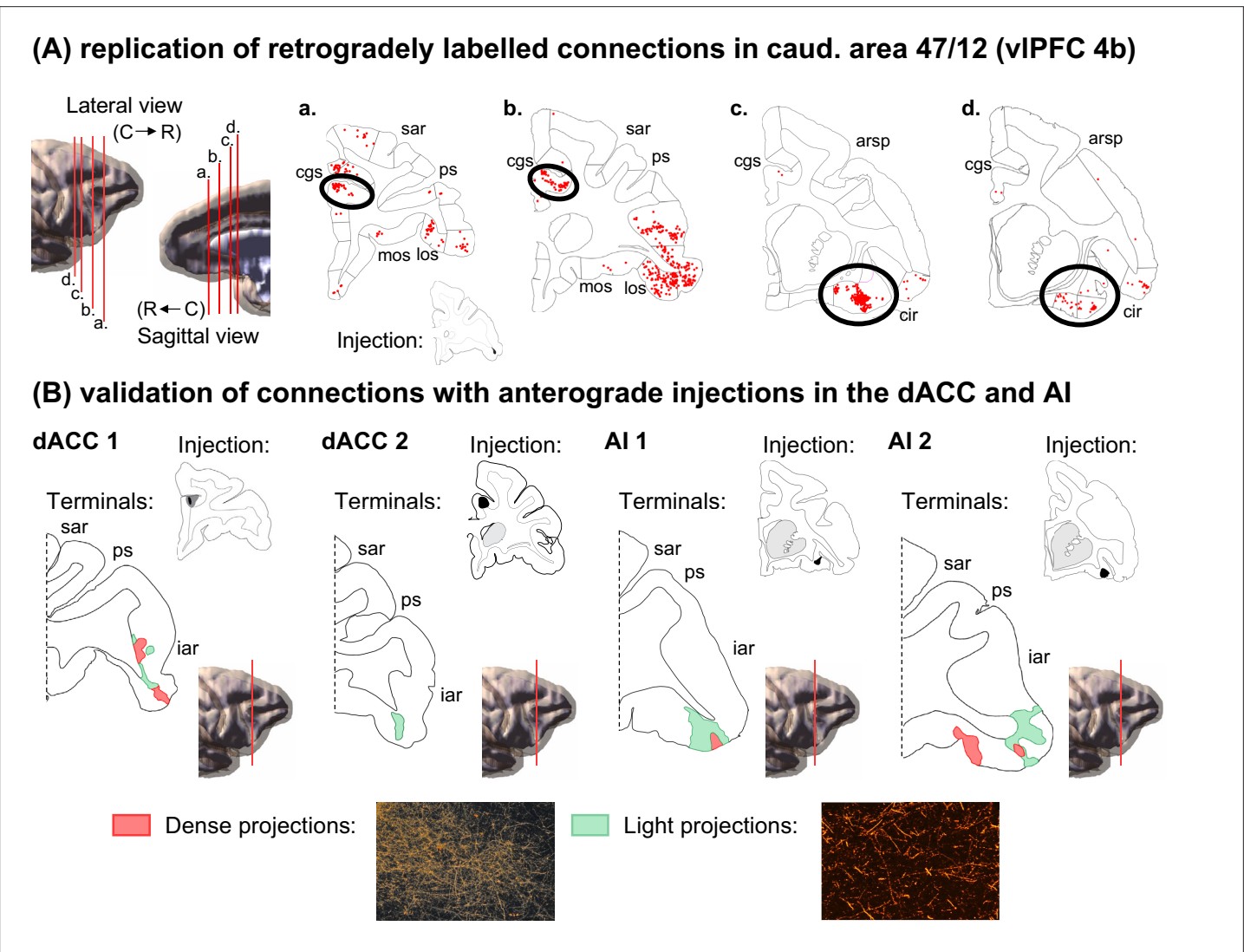

**Figure 3.** Anatomical replication and validation of the caudal 47/12 as a salience network node. (**A**) Coronal sections and the respective labeled cells (red dots) from the validation retrograde tracing injection in caudal 47/12 (case 4b). 3D models represent the location of coronal slices. (**B**) Coronal sections and the respective labeled terminal fields from the validation anterograde tracer injections in the dACC and AI (red areas correspond to dense axonal projections and green areas to light axonal projections). *Abbreviations:* arsp = arcuate sulcus spur; cgs = cingulate sulcus; cir = circular sulcus; iar = inferior arcuate sulcus; los = lateral orbital sulcus; mos = medial orbital sulcus; ps = principal sulcus; sar = superior arcuate sulcus.

The online version of this article includes the following figure supplement(s) for figure 3:

**Figure supplement 1.** Terminal fields from injections in caudal vlPFC area 47/12 within the dorsomedial thalamus (1 x amplification), sublenticular extended amygdala (1 x), and periaqueductal gray (2 x).

region in the macaque brain is enriched with von Economo neurons (*Evrard et al., 2012*; *Figure 2C*), a cell type rare in the brain but characteristic of the SNet (*Seeley et al., 2007*; *Seeley, 2019*). These data demonstrate that a specific vlPFC region, caudal 47/12, is tightly linked to the two SNet nodes. Projection patterns from other cortical areas are shown in *Figure 2—figure supplement 2*.

To replicate these results, we placed an additional retrograde injection at a similar location in caudal 47/12 and found clusters of labeled cells in the same positions within the dACC and AI (*Figure 3A*). Moreover, this injection site was highly correlated with the original caudal 47/12 injection in regards of overall distribution of connectivity strengths across the frontal and insular cortices (rho = 0.70, p<<0.01). To verify the convergence of dACC and AI inputs to the caudal area 47/12, small anterograde tracer injections were placed at the same location as the clusters of dACC and AI-labeled cells (*Figure 3B*). Fibers from these injection sites terminated in the caudal area 47/12. These results are consistent with similar injections within the vlPFC, dACC, and AI reported in qualitative studies (*Petrides and Pandya, 2002*; *Pandya et al., 1981*; *Carmichael and Price, 1996*; *Morecraft et al., 2012*; *Morecraft et al., 2015*; *Mesulam and Mufson, 1982*), and support our findings that there are convergent inputs from the dACC and AI to specific regions of the vlPFC.

The SNet is also characterized by specific subcortical connections, including the SEA, ventral striatum (VS), DT, hypothalamus, SN/VTA, and PAG (*Seeley et al., 2007*; *Uddin, 2016*; *Seeley, 2019*). Importantly, following anterograde injections into the vlPFC, area 44 has light terminal labeling in DT, hypothalamus, and SN/VTA, but not in the SEA and VS. In area 45 terminals were predominantly found in DT, but not in other subcortical nodes. Rostral and mid 47/12 have terminals in DT and SEA. Mid 47/12 also lightly projected to the SN/VTA and lateral hypothalamus. Caudal 47/12 had a particular combination of projections, with dense terminal fields located in the SEA, DT, SN/VTA, hypothalamus, and PAG (*Figure 3—figure supplement 1*). There were fibers and terminals located along the base of the brain streaming through the SEA, with some terminating in the lateral hypothalamus. Moreover, dense terminals fields were also located in the DT, with fewer fibers in the PAG. However, consistent with previous cortico-striatal studies, there were no fibers in the VS. Indeed, vlPFC fibers terminate dorsal to the VS stretching from the ventral rostral putamen and to the central caudate nucleus, just dorsal to the VS (*Gerbella et al., 2016*; *Haber and Knutson, 2010*; *Averbeck et al., 2014*). These connections are consistent with previous anatomical studies (*Giguere and Goldman-Rakic, 1988*; *Stefanacci and Amaral, 2000*; *An et al., 1998*), and provide additional evidence endorsing the role of the caudal area 47/12 in the SNet.

## The SNet node within the caudal area 47/12 can be identified using NHP fcMRI

We then investigated how well these anatomical connectivity patterns may correspond to resting state functional connectivity patterns measured by fMRI. Using data from five macaque monkeys, we placed seven seeds of 3 mm radius in matched locations to our anatomic injection sites and calculated the functional connectivity between each seed and all brain voxels. Masks for the dACC and AI (*Figure 4—figure supplement 1A*) were created with reference to the clusters of cells observed in the retrograde data. Notably, the macaque SNet has a shorter rostrocaudal distribution of the dACC component (*Touroutoglou et al., 2016*) compared to the human SNet (*Seeley et al., 2007*). This distribution was considered during the delineation of the dACC mask. The connectivity strength was computed as the average of absolute connectivity values inside each mask. We also performed $10^6$ random permutations of voxels across the brain volume and computed the random distribution of connectivity strengths in each mask. Importantly, there is high individual variability in the functional organization of the caudal aspect of area 47/12 (*Ren et al., 2021*). Thus, given the limited sample size, the caudal 47/12 seed has a slightly different location for each macaque, although always located within caudal area 47/12. *Figure 4—figure supplement 1B* shows the location of each individual seeds, and the overlapping between them.

The functional connectivity pattern between each vlPFC seed and the dACC mask (*Figure 4A*, top) showed correlations around or below the chance level in rostral and mid area 47/12, areas 45 and 44. Connectivity strength in caudal area 47/12 was above the chance and stands out compared to other brain regions. For functional connectivity between the vlPFC seeds and the AI mask (*Figure 4A*, bottom), again, rostral and mid area 47/12 and area 45 showed connection strengths below the chance level. The caudal area 47/12 presenting the highest connectivity strength among all locations, while

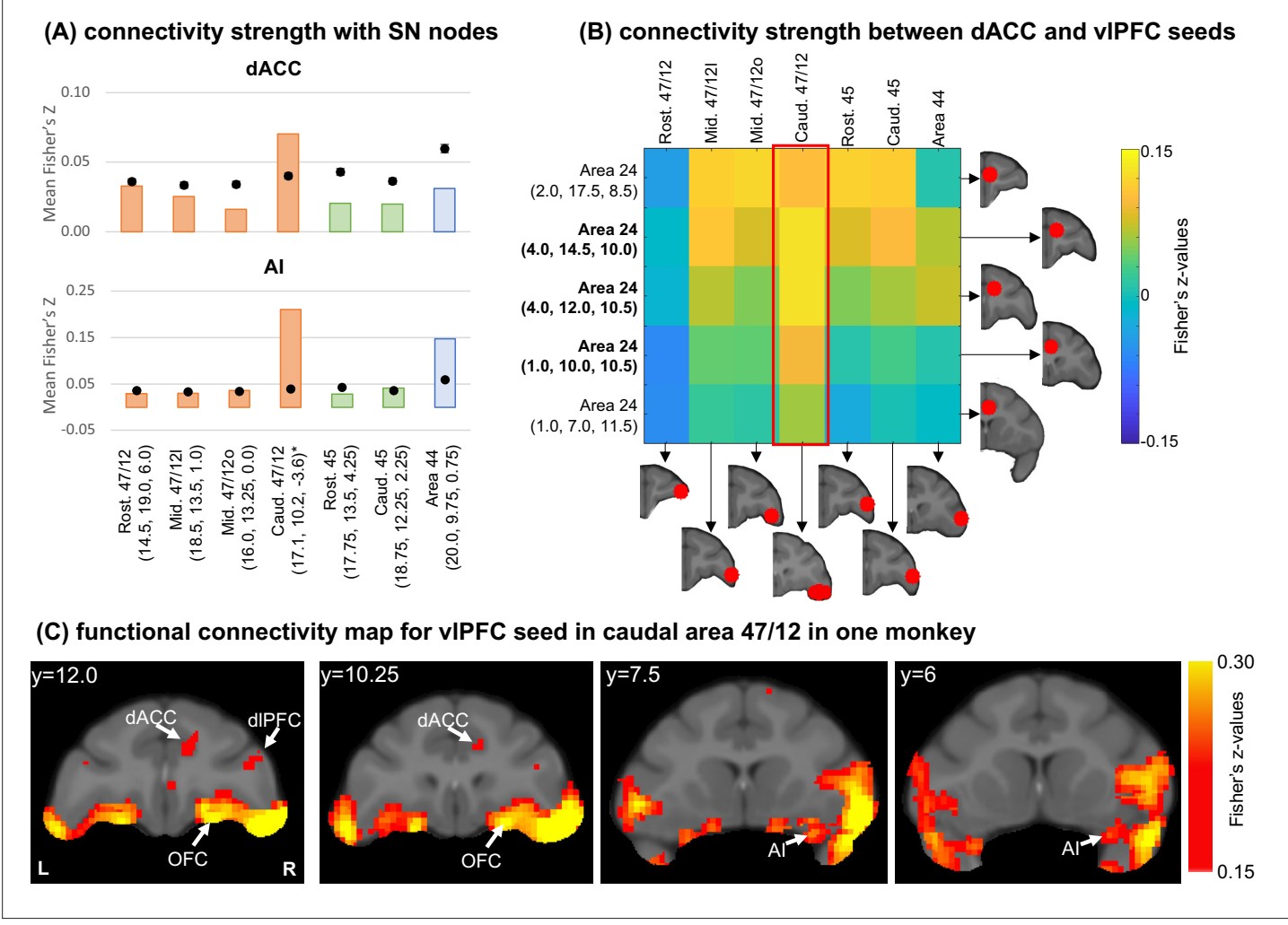

**Figure 4.** Functional connectivity analysis in the macaque brain. (**A**) Average connectivity strength (Fisher's Z-values) between vlPFC seeds and the dACC and AI masks. Orange bars illustrate cases with injections in area 47/12, green bars in area 45, and blue bars in area 44. Black dots show the average and standard-deviation of the voxel permutation analysis. *Centroid's coordinates (please see *Figure 4—figure supplement 1B* for individual seed locations). (**B**) Connectivity strength (Fisher's Z-values) between dACC and vlPFC seeds. In bold the seeds overlapping with the dACC mask. The red frame indicates the connectivity strength between caudal 47/12 and the different dACC seeds. (**C**) Different views of the voxel distribution for the caudal 47/12 seed from one monkey.

The online version of this article includes the following figure supplement(s) for figure 4:

**Figure supplement 1.** Macaque fMRI analysis.

area 44 was also above chance. These connectivity profiles are overall consistent with the anatomical data, with the exception of area 44, which did not show strong connections based on the anatomic tracing (see *Figure 2A*). The results from the fcMRI in area 44 are likely due to the proximity with the caudal area 47/12 and overlap between these seeds.

To ensure the strong connections with the dACC are not artifacts given the proximity of the vlPFC seeds to the AI, we performed a complementary analysis placing 5 seeds within the right dACC (inside and outside the mask). Then, we calculated the seed-to-seed functional connectivity between the dACC and vlPFC (*Figure 4B*). Consistent with the mask analysis, the caudal area 47/12 showed the strongest connections with the dACC seeds within the mask. *Figure 3C* shows the location of voxels within the dACC and AI with high functional connectivity with the seed in caudal area 47/12.

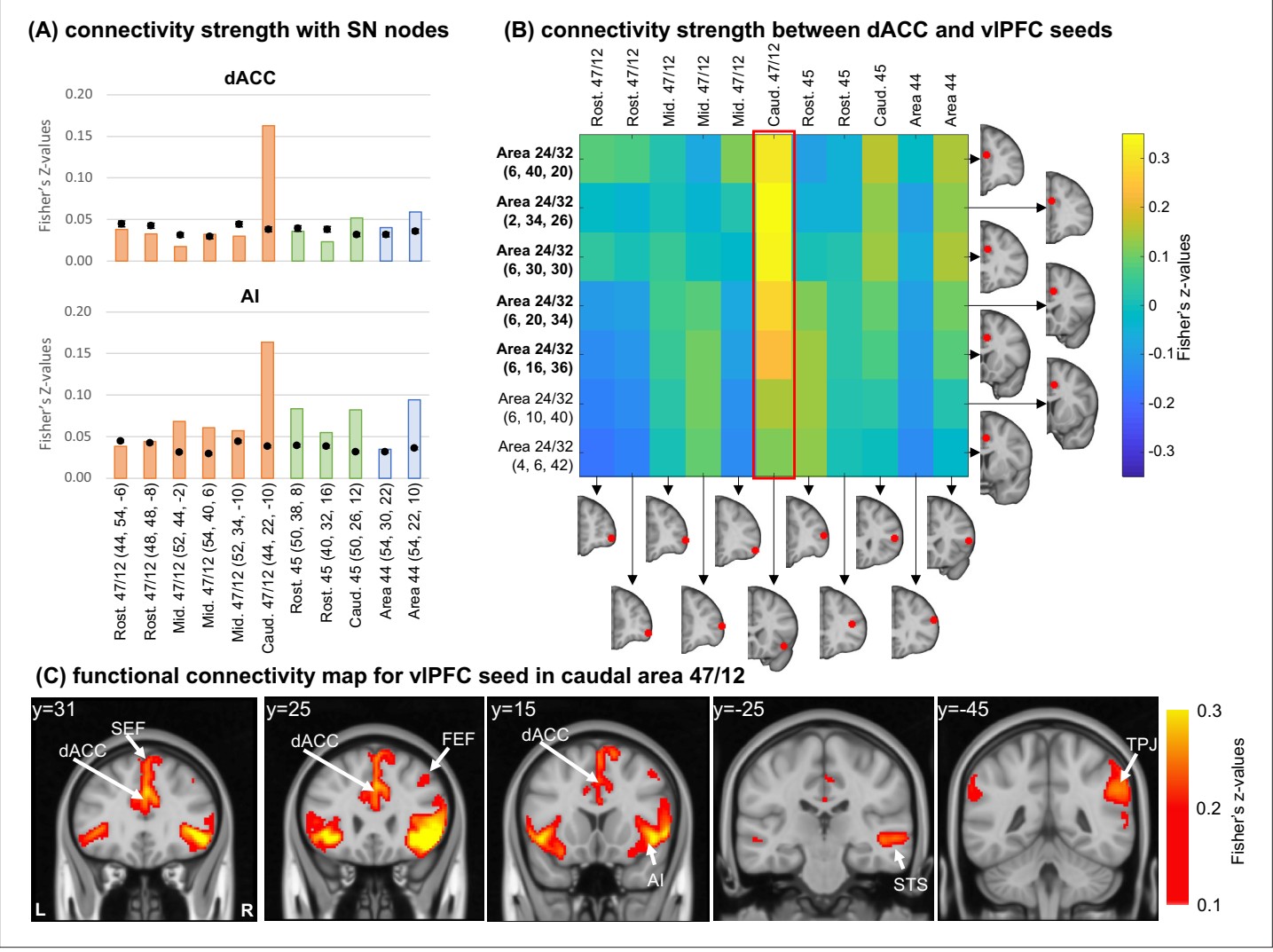

**Figure 5.** Functional connectivity analysis in the human brain. (**A**) Average connectivity strength (Fisher's Z-values) between vlPFC seeds and the dACC and AI masks. Orange bars illustrate cases with injections in area 47/12, green bars in area 45, and blue bars in area 44. Black dots show the average and standard-deviation of the voxel permutation analysis. The red frame indicates the connectivity strength between the caudal 47 seed and the dACC seeds. (**B**) Connectivity strength (Fisher's Z-values) between dACC and vlPFC seeds. In bold the seeds overlapping with the dACC mask. (**C**) Different views of the voxel distribution for the caudal 47/12 seed. All coordinates are in the human MNI space.

The online version of this article includes the following figure supplement(s) for figure 5:

**Figure supplement 1.** Replication of human fMRI analysis.

## A salience network node in the human functional connectivity map of caudal area 47/12

To translate the results from NHP fcMRI to human fcMRI analysis we placed 11 seeds of 5 mm radius across the vlPFC areas. We calculated the functional connectivity between each seed and all brain voxels from 1000 healthy adult subjects from a publicly available, fully preprocessed dataset (Brain Genomics Superstruct Project; *Holmes et al., 2015*). Masks for the dACC and AI (*Figure 5—figure supplement 1A*) were created outlining regions homologous to those containing clusters of cells (*Figure 2B–C*; *Figure 3A*; *Mai et al., 2015*). The computation of connectivity strength and voxel permutation analysis followed the same approach used for the monkey data. Importantly, although the human caudal 47/12 also presents high individual variability (*Ren et al., 2021*), individual seed placement was not necessary. The larger sample size in humans reduced the effect of this variability in our results when using the same seed placement.

Overall connectivity strength with the dACC mask (*Figure 5A*, top) was below, or around, the chance level, with exception of the caudal-most seeds in each vlPFC area. Specifically, the strongest connection was with the caudal area 47/12, similar to the results in the NHP anatomy and imaging data. The connectivity pattern observed between each seed and the AI mask (*Figure 5A*, bottom) is also consistent with the patterns observed in the NHP anatomy. Specifically, connectivity strengths between AI and area 47/12 are organized in a light rostro-caudal gradient, with the strongest connection in caudal area 47/12. This gradient is also consistent with the pattern observed in the NHP tract tracing (*Figure 2A*). Areas 45 and 44 also presented connectivity strengths with the AI and dACC masks around the chance level for the dACC mask and slightly higher for the AI. For all these cases, however, the observed strengths were still lower than the caudal area 47/12. Additionally, we placed 7 seeds within the right dACC (inside and outside the mask) and calculated the seed-to-seed functional connectivity between dACC and vlPFC seeds (*Figure 5B*). As expected, the vlPFC seed in caudal area 47/12 was the one showing the strongest connectivity with the dACC seeds within and around the created mask. These results are consistent with the NHP anatomical and fcMRI data, support the caudal area 47/12 as a node of the SNet. The translation of these nodes and connections across species is supported by anatomic-functional homologies within the vlPFC of humans and NHP (*Petrides and Pandya, 2002*; *Neubert et al., 2014*). We repeated the experiment using smaller (3 mm) and larger (7 mm) seeds to show that these results are independent to the seed size and possible overlapping of the original seed with the beginning of the insular cortex (*Figure 5—figure supplement 1B-C*). Test-retest analysis using two subsamples of 500 subjects also attest for the robustness of the reported results. Finally, we found high functional connectivity between the caudal 47/12 seed and a cluster of voxels in the dACC and AI, two main cortical nodes of the SNet (*Seeley et al., 2007*; *Seeley, 2019*; *Figure 5C*).

## Discussion

### Summary

The presence of salient stimuli activates the SNet and also activates the vlPFC (*Downar et al., 2001*; *Downar et al., 2002*; *Hampshire et al., 2009*; *Hampshire et al., 2010*; *Walther et al., 2011*). However, due to inherent limitations of functional MRI in deciphering signal locations between adjacent cortical areas, the vlPFC component of the SNet is largely ignored, with the assumption that activation is simply part of the AI signal (*Seeley et al., 2007*; *Seeley, 2019*; *Sridharan et al., 2008*). This assumption has had important ramifications for understanding, not only the SNet, but also how the three attention networks might be anatomically linked. In this study, we provide cross-modal and cross-species evidence, based on connectivity, for a separate SNet node located in the caudal area 47/12 within the right vlPFC. This region showed a peak of anatomical and functional connectivity with the main cortical nodes (dACC and AI), as well as anatomical projection to subcortical nodes of the SNet (DT, hypothalamus, SN/VTA, SEA, and PAG). In addition to extending our understanding on the structure of the SNet, our experiment also provides an important methodological contribution to mapping large-scale brain networks. Although fMRI is useful to provide a general view of these circuits, only the precision of NHP tracing is capable of describing the specificities of individual connections, and how they are characterized in each network (*Haber et al., 2020*), as demonstrated here.

### Caudal 47/12 is a node in the SNet

The proposed inclusion of caudal 47/12 in the SNet is primarily based on two lines of anatomic evidence: first, the presence of direct monosynaptic connections to specific regions within the two main cortical SNet nodes, the dACC and AI; and second, a pattern of connections with subcortical areas that are also considered part of the SNet. At the cortico-cortical level, our innovative combination of neuroanatomical tracing methods in NHP with random sampling analysis showed that this area is tightly linked to the dACC and AI. We identified anatomical connectivity strengths significantly above chance levels for each vlPFC subregion and calculated the strength of inputs from the two main cortical nodes of the SNet (dACC and AI). Importantly, we had several injections in area 47/12, which is a particularly large region that can be further subdivided based on connectivity (*Petrides and Pandya, 2002*; *Carmichael and Price, 1995a*; *Borra et al., 2011*; *Saleem et al., 2014*). The peak of connections from both the dACC and AI to the vlPFC specifically targeted the caudal 47/12. In fact,

the strength of the dACC connections was twice as high as connections to other vlPFC subdivisions. Anterograde injections in the dACC corroborated the existence and strength of these connections to caudal 47/12. The cluster of cells from AI projecting to caudal 47/12 was identified predominantly in the rostral portions of the AI. Anterograde injections in this rostral AI region confirmed its connections with caudal 47/12. This AI region is also characterized by the presence of a group of unique neurons (von Economo neurons - VENs), in both humans and NHPs (*Evrard et al., 2012*; *Allman et al., 2010*). VENs have distinctive properties, including fast axonal electric conduction between projected areas (*Allman et al., 2011*), which allows for quick identification of salient stimuli (*Seeley et al., 2007*; *Seeley, 2019*). Importantly, VENs are predominantly found in the right hemisphere compared to the left (*Evrard et al., 2012*; *Allman et al., 2010*), the same hemisphere of the caudal 47/12 SNet node candidate.

The SNet also has subcortical components: the DT, hypothalamus, SN/VTA, SEA, VS, and PAG (*Seeley et al., 2007*; *Uddin, 2016*). We found that most sections of vlPFC displayed partial connectivity to subcortical nodes. Specifically, all areas projected axon terminals to DT, but connections with other subcortical regions varied per vlPFC location. Area 44 projected to the DT, lateral hypothalamus and SN/VTA, but not in the SEA and VS. Area 45 terminals were predominantly found in DT, but not in other subcortical nodes. Mid and caudal 47/12 projected to the DT, lateral hypothalamus, SEA, and SN/VTA, but not to VS. However, caudal 47/12 projections were denser than those observed from mid 47/12. Caudal 47/12 stands out from mid 47/12 given its combination of strong connections with the cortical SNet nodes, and dense projections to the subcortical nodes, providing further support that this location is part of the SNet.

We further translated these tracing results by probing their consistency with fcMRI in NHP (*Haber et al., 2020*). The seed placement in the NHP corresponded to the injection locations. We computed the connectivity strength with two cortical masks created corresponding to the cell clusters in the AI and dACC. As expected, caudal 47/12 showed the highest connectivity strength with both the dACC and AI SNet nodes. These results were replicated when placing seeds within the dACC. However, one potential limitation of our analysis is the existence of a peak of rsFC between the area 44 and the cortical SNet nodes. A possible reason for this result is the spatial overlapping between seeds in caudal 47/12 and area 44, given the resolution of the MRI data available. This resolution limitation highlights the advantages of cross-modality comparisons within the same species when finely delineating brain connectivity to avoid misleading conclusions (*Haber et al., 2020*). Similar patterns of vlPFC connectivity were observed when we systematically placed seeds throughout the human vlPFC. Seeds placed in caudal area 47/12 showed the maximum connectivity strength with both dACC and AI masks, consistent with our anatomical and imaging results in NHP. When placing seeds within the dACC mask for a seed-to-seed analysis, caudal 47/12 again showed the strongest connections with dACC subareas. One important aspect of our results is that, in both NHP and humans, caudal 47/12 connections to subcortical SNet nodes were not as distinguishable as in the tracing data. This limitation is somewhat expected. A previous study using a seed-based rsFC approach to replicate large-scale networks also reported weaker subcortical connections within the SNet (*Buckner et al., 2011*).

This cross-modality and cross-species study provides empirical evidence that caudal area 47/12 is anatomically and functionally connected with the SNet. This location in humans is within the vlPFC area mistakenly merged with the AI into the fronto-insular cortex (FIC) definition (*Seeley et al., 2007*; *Sridharan et al., 2008*). However, caudal area 47/12 and AI are separate structural entities, with different anatomical organization and connectivity profiles (*Petrides and Pandya, 2002*; *Morecraft et al., 2015*; *Mesulam and Mufson, 1982*; *Gerbella et al., 2007*; *Evrard et al., 2014*; *Mufson and Mesulam, 1982*). Together, these data support that caudal 47/12 should be considered as an independent SNet node, separate from the original AI/FIC definition.

## Possible roles of caudal 47/12 within the SNet

In addition to identifying salient stimuli, the SNet recruits behaviorally appropriate responses (*Seeley et al., 2007*; *Menon and Uddin, 2010*). For this purpose, each cortical SNet node has a specific function. The dACC is related to action selection (*Rushworth, 2008*; *Menon, 2015*), given its connections with motor control regions (*Morecraft et al., 2012*). The AI is the node combining sensorial, interoceptive, and limbic information to process salient stimuli (*Menon, 2015*; *Uddin, 2015*) due to its cortico-cortical connections with sensory and limbic regions (*Ongur, 2000*; *Augustine, 1996*).

In addition to projections from AI and dACC, caudal area 47/12 is connected with sensory areas in the temporal pole, cognitive control regions in the PFC, and premotor areas in the frontal cortex (*Petrides and Pandya, 2002*). We propose that the caudal 47/12 node may have two main functions in the SNet. First, caudal 47/12 may predict possible outcomes associated with salient stimuli identified by the AI. One example of stimulus-outcome predictions is the estimation of reward probabilities. Excitotoxic lesions in NHP area 47/12 (including its caudal portion) of macaques impaired choices based on outcome availability after cue presentation (*Rudebeck et al., 2017*). Transient disruption of caudal 47/12o caused by focused ultrasound also led to changes in choice-outcome credit assignment on a probabilistic reversal learning task (*Folloni et al., 2021*). Neurons in a similar location (mid-caudal area 47/12) anticipate and predict information seeking to resolve uncertainty about future rewards and punishments (*Jezzini et al., 2021*). Caudal 47/12 shows high activation during stimulus-outcome updating when varying the visuospatial cues (*Grohn et al., 2020*). Second, caudal 47/12 may be responsible for preparing appropriate behavioral responses later selected by the dACC. Three experiments in NHP show the involvement of the caudal 47/12 in this process. Changes in the grey matter of the caudal 47/12, ACC, and AI, as well as increased functional connectivity between AI and caudal 47/12 are reported when macaques learn object reversal learning tasks (*Sallet et al., 2020*). During win-stay/lose-shift tasks, voxels in the macaque caudal 47/12 show high activation while encoding appropriate decisions (*Chau et al., 2015*). In marmosets, excitotoxic lesions in area 47/12 (including its caudal portion) reduced coping mechanisms to salient negative stimuli (e.g. a fake predator in the experimental environment) (*Agustín-Pavón et al., 2012*; *Shiba et al., 2014*). Moreover, caudal 47/12 connects with other portions of the vlPFC associated with goal-directed movements (*Borra et al., 2011*; *Borra et al., 2017*), which may facilitate the planning of appropriate motor responses.

Studies in humans show that AI is specifically responsible for stimulus processing, and the vlPFC is associated with stimulus-outcome predictions and response preparation. For example, a meta-analysis of stop-signal tasks (SSTs) identified independent activation clusters within the AI and vlPFC (*Cai et al., 2014*). The authors then trained an independent cohort undergoing a new SST and evaluated the fMRI activity in these two clusters. The AI was associated with the identification of salient information (unsuccessful trials), and the vlPFC was responsible for response implementation (inhibitory behaviors) (*Cai et al., 2014*). In a similar experiment, the same research group compared auditory and visual SSTs. Consistent with the first report, the AI was responsive to cue processing while the vlPFC showed a higher role in inhibitory anticipation and implementation (*Cai et al., 2017*). Clinical research also supports the proposed roles of the vlPFC in the SNet. Smokers present abnormal activation in area 47/12 in response to cigarette cues (*de Ruiter et al., 2009*; *Goudriaan et al., 2010*; *Kozink et al., 2010*; *MacLean et al., 2016*). Similar cue-response in the right vlPFC was also reported in gamblers (*de Ruiter et al., 2009*; *Goudriaan et al., 2010*) and patients with eating disorders (*Yokum et al., 2011*). For all these patients, the poor stimulus-outcome estimation may impair response planning (*Zilverstand et al., 2018*). Consequently, they engage in habitual behaviors instead of adequate responses. Addictive behaviors are also related to impaired SNet function (*Zilverstand et al., 2018*). Importantly, altered functional connectivity between the dACC and FIC SNet nodes in these patients is correlated with abnormal vlPFC cue-response (*Janes, 2015*). Altogether, these clinical data provide additional support in favor of the vlPFC functional relevance in the SNet.

## The central role of the vlPFC in attention networks

Here, we demonstrated that the caudal 47/12 is an independent node of the SNet. In addition to the SNet, different subregions of the vlPFC are also physiologically (*Rossi et al., 2007*; *Wardak et al., 2010*; *Kadohisa et al., 2015*; *Bichot et al., 2015*; *Bichot et al., 2019*; *Hartwigsen et al., 2019*; *Buckner et al., 2011*; *Romanski, 2007*; *Kar and DiCarlo, 2021*; *Romanski and Chafee, 2021*) and anatomically (*Petrides and Pandya, 2002*; *Borra et al., 2011*; *Saleem et al., 2014*; *Frey et al., 2014*; *Gerbella et al., 2010*) associated with the main nodes of the VANet (mid and caudal area 47/12) and DANet (areas 44 and 45). Importantly, the caudal area 47/12 (SNet) is highly interconnected with other portions of area 47/12 (VANet) and both areas 44 and 45 (DANet) (*Petrides and Pandya, 2002*; *Borra et al., 2011*; *Saleem et al., 2014*; *Frey et al., 2014*; *Gerbella et al., 2010*). Thus, the three attention networks interface extensively within a vlPFC micro-network. vlPFC's contribution to attention is augmented by the fact that the vlPFC receives input from other areas of the FC. For example, the OFC is tightly linked to the vlPFC (*Petrides and Pandya, 2002*; *Carmichael and Price, 1996*)

and provides relevant information regarding value updating (*Rudebeck et al., 2017*; *Murray and Rudebeck, 2018*). The vlPFC is also closely connected to the dlPFC (*Petrides and Pandya, 2002*; *Petrides and Pandya, 1999*; *Carmichael and Price, 1995b*), supporting executive control functions (*Seeley et al., 2007*). Based on its connectivity profile, we propose the vlPFC as an integrative hub combining high level cognitive processing of attended stimuli and switching between the main attention networks. Specifically, the vlPFC may be the area responsible for bridging the gap between the detection (VANet) and selection (DANet) of relevant stimuli, predicting outcomes, and preparing adequate behavioral responses later coordinated by the SNet. These processes together, explain the critical role of the vlPFC in cognitive and behavioral flexibility (*Dajani and Uddin, 2015*; *Badre and Wagner, 2006*; *Waegeman et al., 2014*).

## Materials and methods

### Injection sites

Ten adult male macaque monkeys (eight *Macaca mulatta*, one *Macaca fascicularis*, and one *Macaca nemestrina*) were used for these tracing studies. All tracer experiments and animal care were approved by the University Committee on Animal Resources at University of Rochester (protocol number UCAR-2008–122 R) and conducted following the National Guide for the Care and Use of Laboratory Animals. Retrograde tracers were injected into the right vlPFC (*Figure 1A*), including one in area 47/12, one in area 47/12o and three in area 47/12 l, two in area 45, one in area 44. Surgical and histological procedures were conducted as previously described (*Haber et al., 2006*; *Heilbronner and Haber, 2014*; *Safadi et al., 2018*; *Tang et al., 2019*). Anterograde tracers were injected into the dACC (two injections, *Figure 1B*) and the FIC (two injections, *Figure 1C*). Stereotaxic coordinates for the injection sites were located using pre-surgery structural MR images. Monkeys received injections of one or more of the following bidirectional tracers: Lucifer Yellow (LY), Fluororuby (FR), or Fluorescein (FS). All tracers were conjugated to dextran amine (Invitrogen) and had similar transport properties (*Rajakumar et al., 1993*).

Twelve to 14 days after the surgery, monkeys were deeply anesthetized and perfused with saline, followed by a 4% paraformaldehyde/1.5% sucrose solution. Brains were post-fixed overnight and cryoprotected in increasing gradients of sucrose (*Haber et al., 2006*). Serial sections of 50 mm were cut on a freezing microtome, and one in every eight free-floating sections was processed to visualize LY, FR and FS tracers, as previously described (*Heilbronner and Haber, 2014*; *Safadi et al., 2018*; *Tang et al., 2019*). Sections were mounted onto gel-coated slides, dehydrated, defatted in xylene overnight, and cover slipped with Permount. In cases in which more than one tracer was injected into a single animal, adjacent sections were processed for each antibody reaction.

### Anatomical tracing analysis

We first divided the FC in 23 areas and the IC in 4 areas based on the atlas by *Paxinos et al., 2000*, in conjunction with detailed anatomical descriptions (*Pandya and Seltzer, 1982*; *Preuss and Goldman-Rakic, 1991*; *Vogt et al., 1995*; *Vogt, 2009*). The rationale for using the atlas of *Paxinos et al., 2000* is the homologous labeling of regions in the macaque and human brains (*Petrides, 1994*; *Petrides et al., 2012*). Then, FC and IC areas were grouped according to common cytoarchitectonic characteristics: area 10 (including subdivisions 10, 10d, 10l, and 10m), 25, 14 (14o and 14m), 11 (11, 11m, and 11l), 13 (13, 13a, 13m, and 13l), 24 (24a, 24b, and 24c), 32, 46 (46v, and 46d), 9 (9l, 9m, 9/32, 9/46, 9/46v, and 9/46d), 8 (8/32, 8a, 8ad, 8av, and 8b), 6m (6/32, and 6m), 6d (6dc/F2, and 6dr/F7), 6v (6vc/F4, 6vr/F5, and ProM), OPAl, OPro, AI, DI, GI, and IPro.

### Retrograde analysis

To evaluate the strength of afferent projections from the FC and IC to the vlPFC, light field microscopy under 20 x objective was used to identify retrogradely labeled cells, as previously described (*Tang et al., 2019*; *Choi et al., 2017*; *Choi, 2017*). StereoInvestigator software (MicroBrightField Bioscience, U.S.A) was used to stereologically count cells in one of every 24 sections (1.2 mm interval). Cell counts were obtained in 19 FC/IC areas previously listed. For each case, the connectivity strength (CS) between each area and the injection site was estimated by a percent score (*Tang et al., 2019*):

$$CS_i = \frac{c_i}{c_{total}} \qquad (1)$$

where $CS_i$ is the connectivity strength for the i-th area, $c_i$ is the cell count in the i-th area, and $c_{total}$ is the total number of labeled cells across all FC/IC areas.

We also performed a random sampling analysis to evaluate the connectivity strengths expected by chance in each area. For this, the total number of cells in each case was randomly assigned to each FC or IC area with a probability given by the volume of the area. The connectivity strength was then calculated according to *Equation 1*. This procedure was repeated $10^6$ times to create a random distribution. The 95% confidence intervals (CI) of these random distributions were computed for each one of the 19 FC/IC areas in each case.

Finally, we calculated the Spearman correlation between the connectivity strength across the 19 FC/IC areas in both caudal 47/12 cases.

## Anterograde analysis

For the dACC and FIC injection cases, dark field light microscopy under 1.6 x, 4 x, and 10 x objectives was used with Neurolucida software (MicroBrightField) to trace outlines of dense or light focal projections to the caudal 47/12. 'Dense projections' were characterized by condensed groups of fibers visible at 1.6 x with discernible boundaries (*Choi et al., 2017*; *Choi, 2017*). Condensed group of fibers where individual terminals could be discerned were labeled as 'light projections' (*Figure 3B*, bottom).

## Functional neuroimaging

### Macaque dataset

The macaque fcMRI maps were generated from five adult monkeys (*Macaca mulatta,* three females, ages 6–7 years, weights 2.5–6.7 kg) from the Nathan Kline Institute. Data from two of these monkeys are publicly available with the NKI dataset (*Xu et al., 2018*) in the PRIMatE Data Exchange (PRIME-DE) consortium (*Milham et al., 2018*). These monkeys had four anesthetized scanning sessions with monocrystalline iron oxide ferumoxytol (MION) as the contrast agent. Each session consists of 4–8 scans (8 min per scan). The NKI Institutional Animal Care and Use Committee (IACUC) protocol approved all imaging methods and procedures in NHP (protocol numbers AP2016-568 and AP2019-642).

### Macaque data acquisition

All MRI data were collected using an 8-channel surface coil adapted for monkey head scanning on a 3.0 Tesla Siemens Tim Trio scanner (Siemens, Erlangen, Germany). Structural images were obtained using a T1-weighted sequence (TR = 2500ms, TI = 1200ms, TE = 3.87ms, FA = 8°, 0.5×0.5 × 0.5 mm voxels). Functional data were collected using a gradient-echo EPI sequence (TR = 2000ms, TE = 16.6ms, FA = 45°, 1.5×1.5 × 2 mm voxels, 32 slices, FOV = 96 × 96 mm). Monocrystalline iron oxide ferumoxytol (MION) solution was injected at iron doses of 10 mg/kg IV before the MRI scanning. The monkey was sedated with an initial dose of atropine (0.05 mg/kg IM), dexdomitor (0.02 mg/kg IM), and ketamine (8 mg/kg IM) intubated and maintained at 0.75% isoflurane anesthesia (inspiration) during the scanning. Respiration and heart rate were measured during all fMRI sessions through Biopac software integrated with the scanner. For additional details on this dataset please refer to the original paper (*Xu et al., 2018*).

### Macaque data preprocessing

Structural data preprocessing included the following steps:1. spatial noise removing and bias field correction using ANTs; 2. brain extraction and segmentation into gray matter, white matter and cerebrospinal fluid using FSL and FreeSurfer; and 3. reconstructing the native white matter and pial surface using FreeSurfer.

Functional data were preprocessed according to the pipeline described in the original paper reporting this NHP dataset (*Xu et al., 2018*). Briefly, we used the following steps: 1. first 5 frames of BOLD data were dropped, constant offset and linear trend over each run were removed; 2. six parameters were obtained by motion correction with a rigid body registration algorithm; 3. spatial smoothing was performed with a Gaussian kernel of FWHM 2 mm; 4. each run was then normalized for global mean signal intensity; 5. a band-pass temporal filter was applied to retain frequencies to 0.01 Hz - 0.1 Hz, and to account for cyclical noise arising from respiratory/cardiovascular apparatus;

6. head motion, whole-brain signal, ventricular and white matter signals were removed through linear regression; 7. the preprocessed fMRI data was then registered to the macaque MNI template and down-sampled to the 1 mm resolution for further analysis.

## Macaque functional connectivity analysis

Seven 3-mm-radius seeds were placed in corresponding locations to our vlPFC injection sites according to the macaque MNI space (*Figure 4A*, *Supplementary file 1A*; *Frey et al., 2011*). The resulting connectivity matrices were later linear projected to the macaque MNI space with 0.25 mm resolution. We created a mask for the dACC, and FIC based on the cluster of cells identified using the retrograde tract-tracing. Then, we used the Fisher r-to-s transformation to correct the correlation values of each voxel, and the functional connectivity strength between each mask and a seed was calculated as the absolute average value within the respective mask.

To evaluate if our results were different from the chance level, we created a random distribution of connectivity strengths in each mask. For this, we performed a random permutation of voxels across the brain volume. Then, we calculated the functional connectivity strength between each mask and seed as previously described. This procedure was repeated $10^6$ times to create a random distribution. The 95% confidence intervals (CI) of these random distributions were computed for each mask in each case.

As a secondary analysis, we placed five 3-mm-radius seeds inside and outside the dACC mask, according to the macaque MNI space (*Figure 4B*, *Supplementary file 1A*; *Frey et al., 2011*). Then, we calculated the functional connectivity between each vlPFC and dACC seeds. Before analysis, these values were also r-to-z transformed.

## Human dataset

For the cross-species functional connectivity analysis, we used a dataset consist of 1,000 young, healthy adult participants (mean age 21.3±3.1 years; 427 males) from the Brain Genomics Superstruct Project (GSP) (*Holmes et al., 2015*). Each participant performed one structural MRI run and 1–2 resting-state fMRI runs (6 min 12 s per run). All participants provided written informed consent following guidelines set by the Institutional Review Boards of Harvard University or Partners Healthcare.

## Human data acquisition

All MRI data were acquired using a 12-channel head coil on matched 3T Tim Trio scanners (Siemens, Erlangen, Germany). Structural data were obtained by a multi-echo T1 weighted gradient-echo image sequence (TR = 2200ms, TI = 1000ms, TE = 1.54ms for image 1 to 7.01ms for image 4, FA = 7°, 1.2×1.2 × 1.2 mm voxels, and FOV = 230). Resting-state functional MRI images were collected using the gradient-echo EPI sequence (TR = 3000ms, TE = 30ms, flip angle = 85°, 3×3 × 3 mm voxels, FOV = 216, and 47 axial slices collected with interleaved acquisition). Participants were instructed to stay awake and keep their eyes open during the scanning.

## Human data preprocessing

Structural MRI data were preprocessed using the 'recon-all' pipeline from FreeSurfer software package. The individual surface mesh was reconstructed and registered to a common spherical coordinate template.

Functional MRI data were processed using a well-stablished preprocessing pipeline for functional connectivity analysis (*Van Dijk et al., 2010*), including: 1. slice timing correction using SPM; 2. head motion correction by FSL; 3. normalization for global mean signal intensity across runs; 4. band-pass filtering (0.01–0.08 Hz); and 5. regression of motion parameters, whole-brain signal, white matter signal, and ventricular signal. The preprocessed fMRI data were then registered to the MNI152 template and downsampled to a 2 mm spatial resolution. Spatial smoothing with a 6 mm FWHM kernel was performed on the fMRI data within the brain mask.

## Human functional connectivity analysis

Eleven 5-mm-radius seeds were placed in corresponding locations to our vlPFC injection sites according to the MNI152 template (*Figure 5A*, *Supplementary file 1B*). After the creation of a dACC and an FIC mask in homologous positions of the cell clusters identified in the macaque retrograde

data, correlation values were r-to-z transformed, and the connectivity strength between each seed and mask was calculated as the absolute average value within the respective mask. The same random permutation approach used for the monkey data was repeated here.

As a secondary analysis, we placed seven 5-mm-radius seeds inside and outside the dACC mask, according to the macaque MNI152 template (*Figure 5B*, *Supplementary file 1B*) and calculated the seed-to-seed connectivity between the vlPFC and dACC. The Fisher r-to-z transformation was also applied to these results. Finally, we repeated both analyses using 3-mm and 7-mm-radius seeds to ensure that our results were not driven by the seed size (*Figure 5—figure supplement 1*).

## Acknowledgements

Funding: LRT, JFL, HL, and SNH were supported by the National Institutes of Health (Grant Nos. MH106435 and MH045573). XP and HL received support from the National Natural Science Foundation of China (Grant No. 81790652). GL, BER, and CES received support from the National Institute of Health (Grant Nos. MH111439 and MH109429).

## Additional information

### Funding

| Funder | Grant reference number | Author |
| --- | --- | --- |
| National Institutes of Health | MH106435 | Suzanne N Haber |
| National Institutes of Health | MH045573 | Suzanne N Haber |
| National Natural Science Foundation of China | 81790652 | Hesheng Liu |
| National Institutes of Health | MH111439 | Charles E Schroeder |
| National Institutes of Health | MH109429 | Charles E Schroeder |

The funders had no role in study design, data collection and interpretation, or the decision to submit the work for publication.

### Author contributions

Lucas R Trambaiolli, Conceptualization, Data curation, Formal analysis, Investigation, Methodology, Visualization, Writing – original draft, Writing – review and editing; Xiaolong Peng, Data curation, Formal analysis, Investigation, Methodology, Visualization, Writing – review and editing; Julia F Lehman, Data curation, Investigation, Visualization; Gary Linn, Brian E Russ, Charles E Schroeder, Data curation, Methodology, Validation; Hesheng Liu, Data curation, Methodology, Writing – review and editing; Suzanne N Haber, Conceptualization, Data curation, Funding acquisition, Investigation, Methodology, Project administration, Resources, Supervision, Writing – original draft, Writing – review and editing

### Author ORCIDs

Lucas R Trambaiolli ⓘ http://orcid.org/0000-0001-7824-1929
Xiaolong Peng ⓘ http://orcid.org/0000-0002-4488-9628
Suzanne N Haber ⓘ http://orcid.org/0000-0002-5237-1941

### Ethics

All tracer experiments and animal care were approved by the University Committee on Animal Resources at University of Rochester (protocol number UCAR-2008-122R). The NKI Institutional Animal Care and Use Committee (IACUC) protocol approved all imaging methods and procedures in NHP (protocol numbers AP2016-568 and AP2019-642). All experiments were conducted following the National Guide for the Care and Use of Laboratory Animals.

#### Decision letter and Author response

Decision letter https://doi.org/10.7554/eLife.76334.sa1
Author response https://doi.org/10.7554/eLife.76334.sa2

## Additional files

#### Supplementary files
• Supplementary file 1. ROI centers for fMRI analysis.

• Transparent reporting form

#### Data availability

All anatomical data analysed during this study are included in the manuscript and supporting files. Functional connectivity analyses utilized publicly available datasets: PRIME-DE: https://fcon_1000. projects.nitrc.org/indi/indiPRIME.html GSP: https://www.nature.com/articles/sdata201531.

The following previously published datasets were used:

| Author(s) | Year | Dataset title | Dataset URL | Database and Identifier |
|---|---|---|---|---|
| Homes et al | 2015 | Brain Genomics Superstruct Project initial data release with structural, functional, and behavioral measures | https://www.nature. com/articles/ sdata201531 | Brain Genomics Superstruct Project, 201531 |
| Milham et al | 2020 | PRIMatE Data Exchange (PRIME-DE) | https://fcon_1000. projects.nitrc.org/ indi/indiPRIME.html | PRIMatE Data Exchange (PRIME-DE), PRIME-DE |

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
