## [Editor Report]

This is an interesting quantitative study of the anatomical connections of a region of prefrontal cortex that has often been overlooked because it is at the border of what is typically called ventrolateral prefrontal cortex and orbitofrontal prefrontal cortex. Sometimes it is included as part of ventrolateral prefrontal cortex, sometimes as part of orbitofrontal cortex and sometimes it is simply given little attention because ventrolateral studies focus on the inferior convexity and orbital studies focus on the region between the orbitofrontal sulci. The idea that this is a special region that is different from both the rest of ventrolateral prefrontal cortex and probably the rest of orbitofrontal cortex is important because it helps us understand some otherwise puzzling results. The quantitative analysis of connections was an unusual strength of the study as was the comparison of tracer data in macaques, fMRI connectivity data in macaques, and human fMRI connectivity data.

---

## [Decision Letter]

**Decision letter after peer review:**

Thank you for submitting your article "Anatomical and functional connectivity support the existence of a salience network node within the caudal ventrolateral prefrontal cortex" for consideration by *eLife*. Your article has been reviewed by 3 peer reviewers, including Birte U Forstmann and the Reviewing Editor and Reviewer #1, and the evaluation has been overseen by Chris Baker as the Senior Editor.

Essential revisions:

Overall, this manuscript is well written, interesting, timely and will help resolve the debate in the field. Despite the importance of the work and generally solid methods, there are a number of issues with the current work that need to be addressed. Most notably some additional analyses are required to ensure the validity and reliability of the results that are reported, especially in the macaque imaging data where analyses appear to have been conducted on a high-quality dataset from a single macaque.

We have the following suggestions to improve the manuscript:

1. A major concern is the reliability of the resting-state functional connectivity MRI (fcMRI) analyses. Only one NHP dataset was analyzed while data from 1000 healthy young volunteers were included. What is the reason for including only one NHP dataset and to what extent do the authors believe that this single NHP dataset is representative?

2. Along these lines, the tract-tracing study only includes male NHP's while the fcMRI study includes one female NHP. Why?

3. The figures are generally excellent but could be a little more informative. In each figure showing coronal slices please also include a representation on a sagittal plane of where those coronal slices are in both human and macaques. Similarly, the authors should consider moving parts of figure S1 showing lateral views of the brain and location of injections to the main manuscript figures.

4. The dACC component of the SN usually refers to a more caudal component of the cingulate cortex compared to the one indicated here that appears to be located in a more anterior area (Seeley et al., 2007; Menon et al., 2015). Is there a reason why such anterior portion was chosen? If the part of the ACC involved in the SN in macaques is located more anteriorly in the cingulate cortex as compared to humans, please include reference and provide some discussion on this potential difference. Would the vlPFC-dACC result stand even in more dorsal-caudal portions of dACC? Based on this comment, I wonder if it might be helpful to show the density of labelled neurons in ACC on a medial view that projects to the vlPFC. This way a reader can get a better understanding of the distribution of neurons projecting as area 24 is of considerable length in the A-P plane.

5. At several points throughout the manuscript, the authors highlight the centrality of area 47/12l as the part of vlPFC most involved in the SN. However, I am not fully convinced by the uniqueness of 47/12l over other parts of 47/12 as in multiple figures as it seems as if the results extend into 47/12o (Figure 2, Figure 3, Figure 4, Suppl. Figure 3). Unless the authors can provide additional information/clarification here, I wonder if it might be more appropriate/correct to stick with a broader "caudal 47/12" definition which includes both the lateral and orbital portions. Indeed, the seed-ROI labelled as area 47/12o also seems to be more positively functionally coupled with cingulate and insular ROIs in Figure 3.

6. The analyses of macaque resting state data are from a single animal whereas the data in humans are from 1000 individuals. While there is nothing wrong with this per se and the macaque data are of a very high quality, it raises a question about the reproducibility of the findings in macaques. So, I think that I'm going to need a little more convincing that there is strong functional resting-state connectivity between vlPFC, ACC, and AI in macaques. When the same analyses are conducted in additional animals are the same patterns of functional connectivity observed? To be clear, I'm not asking for more than a few more examples and I believe that there are sufficient open source datasets available for this to be done.

7. On a related note, for the human data there is no indication of the variability of the functional connections between subjects. With such a large sample, the authors actually have the chance to really dig into the data and look at how reproducible the effects are between subjects. I'm not asking for them to run each subject individually, but spitting the sample into smaller groups and testing for robustness would be a good approach to test the reproducibility of the findings.

8. Related to this, the authors state that for the human analyses of functional connectivity, that "Masks for the dACC and AI were created outlining regions homologous to those containing clusters of cells (Figure 1B-C; Figure 2A) (41)". Because these are non-standard masks, it would be helpful if the authors could show the location of these in either the main or supplementary figures.

9. In Figure 3B, I'm interested in why the authors think the functional coupling of the "rostral 47/12" seed is so negatively coupled with cingulate seeds? This negative coupling can be observed for other 47 seeds in the human analysis as well and it is quite in contrast to the positive coupling of other 47/12 seeds. So, I'm going to need a little more information here: (1) Do the authors think that this is a real effect or an artifact of their analyses? (2) When they look at the tract tracing data in macaques, is there a marked difference between anterior and posterior vlPFC?

10. In Figure 4, the ROI labels should match the different cytoarchitectonic portions of the areas they report as reported in Figure 3. At the moment, areas 44, 45 and 47 ROIs just report a code (i.e. 47-01, 47-02, 47-03, 47-04, etc…).

11. The authors use 3 mm seeds in the macaque resting-state analysis and 5 mm seeds in the human. However, the rationale of the choice of the size is not straightforward. I am particularly curious to know why this choice was made, especially considering that the 5 mm seed in the human brain does not seem to cover an equal amount of cortex as the 3 mm one in the macaque when scaling for the overall size of the brain in the two species. The authors do run a second control analysis in the human brain using 3 mm seeds. This is great, but if anything, I feel they should use a seed larger than the 5 mm initially used in order to run an analysis more comparable to the monkey. If the authors were to repeat the analyses in humans with a 7mm or larger seed, what is the result?

12. A recent paper from the Rushworth group (Folloni et al., 2021 Science Advances) has shown that disrupting area 47/12 using focused ultrasound differentially impacts functions arising from the other two areas (ACC and AI) that are part of the same Salience Network. Including reference to this manuscript could be a useful way to prove the functional basis for this circuit connecting vlPFC, ACC and AI.

*Reviewer #1 (Recommendations for the authors):*

Overall, the topic and methodology is very interesting while the tract-tracing results are most compelling. However, there are several major concerns the authors need to address. These are listed below.

1. A major concern is the reliability of the resting-state functional connectivity MRI (fcMRI) analyses. Only one NHP dataset was analyzed while data from 1000 healthy young volunteers were included. What is the reason for including only one NHP dataset and to what extent do the authors believe that this single NHP dataset is representative?

2. Along these lines, the tract-tracing study only includes male NHP's while the fcMRI study includes one female NHP. Why?

In sum, I would suggest that the authors include more NHP fcMRI datasets along with a power analysis.

---

## [Author Response]

Essential revisions:Overall, this manuscript is well written, interesting, timely and will help resolve the debate in the field. Despite the importance of the work and generally solid methods, there are a number of issues with the current work that need to be addressed. Most notably some additional analyses are required to ensure the validity and reliability of the results that are reported, especially in the macaque imaging data where analyses appear to have been conducted on a high-quality dataset from a single macaque.We have the following suggestions to improve the manuscript:1. A major concern is the reliability of the resting-state functional connectivity MRI (fcMRI) analyses. Only one NHP dataset was analyzed while data from 1000 healthy young volunteers were included. What is the reason for including only one NHP dataset and to what extent do the authors believe that this single NHP dataset is representative?

We appreciate the reviewer’s concern. We have included data from four additional monkeys (for a total of five macaques) and updated our results accordingly. We believe that the additional data support the robustness of our results.

2. Along these lines, the tract-tracing study only includes male NHP's while the fcMRI study includes one female NHP. Why?

The tract-tracer cases included in this study are part of the permanent collection of Haber lab, which is limited to male monkeys. The inclusion of new female monkeys is limited by current shortages in female monkeys for research. For the NHP fMRI analysis, we originally used data from one female, but have included additional animals and consequently balancing our sample (two males and three females). As the results significantly sustained our original results, we believe that potential sex differences are not relevant to our findings. Additionally, the human fMRI analysis included a balanced sample (427 males and 573 females) removing the impact of potential sex confounders.

3. The figures are generally excellent but could be a little more informative. In each figure showing coronal slices please also include a representation on a sagittal plane of where those coronal slices are in both human and macaques. Similarly, the authors should consider moving parts of figure S1 showing lateral views of the brain and location of injections to the main manuscript figures.

We moved Figure S1 to main text and added representations in lateral/sagittal views for the locations of coronal slices showed in new Figures 2 and 3.

4. The dACC component of the SN usually refers to a more caudal component of the cingulate cortex compared to the one indicated here that appears to be located in a more anterior area (Seeley et al., 2007; Menon et al., 2015). Is there a reason why such anterior portion was chosen? If the part of the ACC involved in the SN in macaques is located more anteriorly in the cingulate cortex as compared to humans, please include reference and provide some discussion on this potential difference. Would the vlPFC-dACC result stand even in more dorsal-caudal portions of dACC? Based on this comment, I wonder if it might be helpful to show the density of labelled neurons in ACC on a medial view that projects to the vlPFC. This way a reader can get a better understanding of the distribution of neurons projecting as area 24 is of considerable length in the A-P plane.

We appreciate the reviewer’s concern about the A-P distribution of the ACC component of the Salience Network. In both humans and NHP, the ACC component starts in the pregenual ACC and extends caudally to the dACC. A qualitative comparison of our result in tracer data and fcMRI with previous reports in NHP (Touroutoglou et al., 2016, Figure 2) and the human (Seeley et al., 2007, Figure 2) show a similar pattern of cells and activation the A-P distribution.

As suggested by the reviewer, we have discussed these species-specific differences the main manuscript in the Results section.

5. At several points throughout the manuscript, the authors highlight the centrality of area 47/12l as the part of vlPFC most involved in the SN. However, I am not fully convinced by the uniqueness of 47/12l over other parts of 47/12 as in multiple figures as it seems as if the results extend into 47/12o (Figure 2, Figure 3, Figure 4, Suppl. Figure 3). Unless the authors can provide additional information/clarification here, I wonder if it might be more appropriate/correct to stick with a broader "caudal 47/12" definition which includes both the lateral and orbital portions. Indeed, the seed-ROI labelled as area 47/12o also seems to be more positively functionally coupled with cingulate and insular ROIs in Figure 3.

We agree with the reviewers that our caudal injection in area 47/12 is in the border of areas 47/12l and 47/12o. As suggested, we adapted the manuscript to use the broader term “caudal 47/12”.

6. The analyses of macaque resting state data are from a single animal whereas the data in humans are from 1000 individuals. While there is nothing wrong with this per se and the macaque data are of a very high quality, it raises a question about the reproducibility of the findings in macaques. So, I think that I'm going to need a little more convincing that there is strong functional resting-state connectivity between vlPFC, ACC, and AI in macaques. When the same analyses are conducted in additional animals are the same patterns of functional connectivity observed? To be clear, I'm not asking for more than a few more examples and I believe that there are sufficient open source datasets available for this to be done.

As suggested, we have included data from four additional monkeys and updated these results in Figure 4. These new data are consistent with the results using one single animal and support the robustness of our findings.

7. On a related note, for the human data there is no indication of the variability of the functional connections between subjects. With such a large sample, the authors actually have the chance to really dig into the data and look at how reproducible the effects are between subjects. I'm not asking for them to run each subject individually, but spitting the sample into smaller groups and testing for robustness would be a good approach to test the reproducibility of the findings.

As suggested, we performed a test-retest reliability analysis by splitting our sample in two independent subsamples of 500 subjects. As shown in the new Figure 5. – Figure Supp. 1-D, the connectivity matrices between vlPFC-ACC seeds presented similar patterns for both subsamples and significant Spearman’s correlations with the main findings reported in Figure 5-B. This supports the robustness of our findings.

8. Related to this, the authors state that for the human analyses of functional connectivity, that "Masks for the dACC and AI were created outlining regions homologous to those containing clusters of cells (Figure 1B-C; Figure 2A) (41)". Because these are non-standard masks, it would be helpful if the authors could show the location of these in either the main or supplementary figures.

We have included the masks in the new Figure 4. – Figure Supp. 1A (NHP) and Figure 5. – Figure Supp. 1A (humans).

9. In Figure 3B, I'm interested in why the authors think the functional coupling of the "rostral 47/12" seed is so negatively coupled with cingulate seeds? This negative coupling can be observed for other 47 seeds in the human analysis as well and it is quite in contrast to the positive coupling of other 47/12 seeds. So, I'm going to need a little more information here: (1) Do the authors think that this is a real effect or an artifact of their analyses? (2) When they look at the tract tracing data in macaques, is there a marked difference between anterior and posterior vlPFC?

This is an excellent question. Anticorrelations can indicate the existence of connections between two areas, but some preprocessing methods (such as the global signal regression -GSR – used in this study) may introduce spurious anticorrelations in static functional connectivity analyses. Although there is still no consensus about whether GSR should or should not be included in resting state data processing, a recent study showed that GSR is the only denoising method that effectively removes global signals, including artefactual signals and, crucially, global neural activity (Power JD, M Plitt, et al., 2017). Since we opted to use GSR in our preprocessing pipeline to control for global neural activity (e.g., controlling for levels of alertness), we also opted to not discuss or interpret results that could be driven by such influence (e.g., the negative coupling in rostral area 47/12).

10. In Figure 4, the ROI labels should match the different cytoarchitectonic portions of the areas they report as reported in Figure 3. At the moment, areas 44, 45 and 47 ROIs just report a code (i.e. 47-01, 47-02, 47-03, 47-04, etc…).

We agree and have edited the new Figure 5 (and the Figure 5. – Figure Supp. 1) so that the results in humans follow the cytoarchitectonic nomenclature used in the rest of the paper.

11. The authors use 3 mm seeds in the macaque resting-state analysis and 5 mm seeds in the human. However, the rationale of the choice of the size is not straightforward. I am particularly curious to know why this choice was made, especially considering that the 5 mm seed in the human brain does not seem to cover an equal amount of cortex as the 3 mm one in the macaque when scaling for the overall size of the brain in the two species. The authors do run a second control analysis in the human brain using 3 mm seeds. This is great, but if anything, I feel they should use a seed larger than the 5 mm initially used in order to run an analysis more comparable to the monkey. If the authors were to repeat the analyses in humans with a 7mm or larger seed, what is the result?

As suggested by the reviewer, we performed a new analysis using 7mm seeds. As shown in the new Figure 5. – Figure Supp. 1-C, the connectivity matrices between vlPFC-ACC seeds presented significant Spearman’s correlations with the main findings reported in new Figure 5-B using 5mm seeds. Similar correlation values are obtained with smaller seeds (3mm) as shown in Figure 5. – Figure Supp. 1-B. This suggests that the results in human fcMRI are robust independently of the seed size.

12. A recent paper from the Rushworth group (Folloni et al., 2021 Science Advances) has shown that disrupting area 47/12 using focused ultrasound differentially impacts functions arising from the other two areas (ACC and AI) that are part of the same Salience Network. Including reference to this manuscript could be a useful way to prove the functional basis for this circuit connecting vlPFC, ACC and AI.

We appreciate this suggestion. We have included this study in our discussion.